# Clinical EEG of Rett Syndrome: Group Analysis Supplemented with Longitudinal Case Report

**DOI:** 10.3390/jpm12121973

**Published:** 2022-11-29

**Authors:** Galina Portnova, Anastasia Neklyudova, Victoria Voinova, Olga Sysoeva

**Affiliations:** 1Institute of Higher Nervous Activity and Neurophysiology, Russian Academy of Sciences, Moscow 117485, Russia; 2Veltischev Research and Clinical Institute for Pediatrics of the Pirogov, Russian National Research Medical University, Ministry of Health of Russian Federation, Moscow 125412, Russia

**Keywords:** EEG, Rett syndrome, MECP2, background slowing, sensorimotor rhythm

## Abstract

Rett syndrome (RTT), a severe neurodevelopmental disorder caused by MECP2 gene abnormalities, is characterized by atypical EEG activity, and its detailed examination is lacking. We combined the comparison of one-time eyes open EEG resting state activity from 32 girls with RTT and their 41 typically developing peers (age 2–16 years old) with longitudinal following of one girl with RTT to reveal EEG parameters which correspond to the RTT progression. Traditional measures, such as epileptiform abnormalities, generalized background activity, beta activity and the sensorimotor rhythm, were supplemented by a new frequency rate index measured as the ratio between high- and low-frequency power of sensorimotor rhythm. Almost all studied EEG parameters differentiated the groups; however, only the elevated generalized background slowing and decrease in our newly introduced frequency rate index which reflects attenuation in the proportion of the upper band of sensorimotor rhythm in RTT showed significant relation with RTT progression both in longitudinal case and group analysis. Moreover, only this novel index was linked to the breathing irregularities RTT symptom. The percentage of epileptiform activity was unrelated to RTT severity, confirming previous studies. Thus, resting EEG can provide information about the pathophysiological changes caused by MECP2 abnormalities and disease progression.

## 1. Introduction

Rett syndrome (RTT) is a progressive neurodevelopmental disorder with a prevalence ranging from 1:10,000 to 1:20,000 [1]. Different types of MECP2 gene mutations cause RTT. The protein product of this gene (MECP2 protein) controls gene transcription [2,3]. The main characteristics of RTT are abnormal language and psychomotor development, autistic behaviors, breath irregularities, abnormal gait and hand wringing [4,5]. The syndrome follows four different stages: (I) prenatal and early development which is considered to be normal; (II) developmental regression with loss of previously acquired motor and language skills, starting at the age of 6–18 months; (III) pseudostationary period with some communicative and cognitive improvement, which may last for years to decades; (IV) late motor deterioration with progressive disability, which might not occur with a proper therapy [6].

Epilepsy and epileptiform activity often accompany Rett syndrome, being reported in 50–90% of cases [7]. Patients with RTT possess different variants of epileptiform activity including generalized slow spike–wave activity; short-duration spikes, predominantly of centrotemporal origin; and other multifocal epileptiform activity [8,9]. The average age of seizure onset is 4 years, but different types of epileptiform activity might be present much earlier. While the age of seizure onset is linked with the severity of epilepsy, its contribution to the core RTT symptoms is still under debate [10,11]. For example, at the regression stage II, many patients with RTT do not have seizures.

Among other EEG abnormalities often reported in RTT is a general slowing of background EEG [12,13,14,15]. This clinical observation is supported by a few quantitative experimental EEG studies that reported an increase in the power of delta and theta oscillation in RTT [16,17,18]. However, the relation of this generalized EEG slowing to the RTT symptoms is also not very clear.

Sensorimotor rhythms were also a focus of some RTT research considering their relation to motor function, imitation and cognitive control [19,20,21,22]. Typical functional reactivity to passive/active movements and topographical maximum at central scalp electrodes was reported in patients with RTT. At the same time, it had an abnormally low frequency, although no statistical assessment has been done [18,21].

Our study aims to combine clinical, qualitative and experimental/quantitative approaches to the resting EEG analysis in a search for the EEG characteristics that are linked with RTT symptoms. To achieve this goal, we performed a longitudinal assessment of one girl with RTT from 2 to 7 years of age through the disease progression as well as conducted an association analysis between the manifestation of different RTT symptoms and EEG abnormalities in a group of girls with RTT that was additionally compared with their typically developing peers.

## 2. Materials and Methods

### 2.1. Study Design

We analyzed 14 EEG recordings of patient U made between 2015 and 2021 and EEG recordings of 32 patients with Rett syndrome made between 2020 and 2021. Data were collected retrospectively from a review of the medical records and EEG reports. Inclusion criteria were clinically confirmed diagnosis of Rett syndrome as well as genetic abnormalities in the MECP2 gene. Our longitudinal case was chosen based on the number of EEG recordings available, as well as its representative phenotype (see below) and genotype (one of the most common mutations being R255X).

Additionally, the sample of 32 Rett patients was compared with typically developed (TD) children who had no neurological or psychiatric disorders. They were recruited specifically as a control group for this study from an advertisement in the local community.

Parents or legal representatives gave written informed consent to the children’s participation in the study. Children who were able to communicate also provided verbal consent to participate. The research procedure was approved by the ethical committees of the Institute of Higher Nervous Activity and Neurophysiology, Russian Academy of Sciences (protocol No. 2 from 30 April 2020). All aspects of the research conformed to the tenets of the Declaration of Helsinki. 

### 2.2. EEG Registration

EEG data from 32 patients and 41 control subjects were recorded using a 28-channel NeuroTravel (Firenze, Italy) system with connected earlobe electrodes used as a reference and the grounding electrode placed centrally. Electrodes were arranged according to the international 10–10 system. EEG registration was conducted in awake patients with open eyes during the daytime and lasted for 500 s. The signal was sampled at 500 Hz and filtered with an online bandpass filter of 0.016–70 Hz and with a notch filter at 50 Hz. The electrode impedances were below 10 kΩ.

Recordings in patient U were made also during eyes open condition during daytime. Different EEG instruments, such as 19-channel Grass Technology (from 1st to 5th recordings), BiosemiActiveTwo 64-electrode array (6th–8th) and a 28-channel NeuroTravel system (9th to 14th recordings), were used.

### 2.3. Clinical Data

Clinical information was collected from a review of inpatient medical notes, imaging studies and other clinical reports. Data recorded included baseline demographic data (age), prior history of disorders including epilepsy and suspected clinical seizures.

Rett symptom severity was assessed for each patient using the Rett Syndrome Severity Scale (RSS), as modified by [23]. This clinician-rated scale represents an aggregate measure of the severity of clinical symptoms, including motor function (ability to walk and hand use), seizures, respiratory irregularities, ambulation, scoliosis, speech and sleep quality. Each item is scored from 0 (absent/normal) to 3 (severe).

### 2.4. EEG Analysis

EEG periods of 60–500 s of eyes open condition were analyzed for each participant. Independent component analysis (ICA) was used when needed to subtract the most evident artifacts [24]. The three separate neurologists (including Portnova G., MD, PhD) with expert certification identified and interpreted EEG data, reaching common decisions.

The following factors were assessed from EEG recordings:

Epileptiform and paroxysmal activity
Benign sporadic wave discharges, spikes, multi-spikes, classified as benign focal epileptiform discharge of childhood (BFEDC) [25], which were not accompanied by clinical events. The index of this activity was calculated in relation to the analyzed EEG fragments (see Methods); their duration was, on average, from 100 to 900 ms. The topography of this activity was taken into account. Benign variants were included for several reasons. First, we believe that epileptiform activity can be easily misclassified, especially in children with atypical development [26,27,28]. Second, the EEG abnormalities including benign variants of epileptiform activity could be a sign of both brain immaturity and brain pathology [27] and we suggested that we need to take into account this factor when comparing RTT children with their typical peers.Episodic peak–wave or slow spike–wave complexes, which were not accompanied by clinical events and did not show repetitive structure, generalization or secondary generalization and did not exceed the duration of 2 s which are also not accompanied by clinical events and are not systemic in nature. These complexes, although sporadic, may potentially progress to hemi-generalized epileptiform activity. The topography of this activity was also taken into account.Typical or atypical epileptiform discharges manifesting with secondary generalized spike–slow wave discharges or spike–wave discharges. We analyzed accompanying clinical seizures or other clinical events, the duration of epileptiform discharges, the presence of generalization or secondary generalization, topography and complications.

Thus, we considered the following abnormalities: sporadic epileptiform discharges (spikes or sharp waves), periodic discharges (lateralized, generalized or bilateral independent) and electrographic seizures. In addition, we calculated the index of epileptiform activity (%) measured as the summarized time of EEG abnormalities to the time of the whole EEG fragment.

2.General slowing

We registered two types of slowing: generalized background slowing (i.e., general slowing (GS)) and focal slowing in sensorimotor areas [29]. GS met the following inclusion criteria: should be symmetric with no significant difference in amplitudes on laterality and should be registered during more than 50% of EEG fragment duration. We calculated the mean amplitude (GS_Amp) and frequency (GS_Fr), averaged over all electrodes.

3.Beta rhythm

We calculated the average amplitude of the beta rhythm (Beta_Amp) 14–30 Hz over all channels during the whole artifact-free EEG fragment.

4.The sensorimotor rhythm and its focal slowing

The sensorimotor rhythm (SM) referred to oscillations of 3–12 Hz recorded over the sensorimotor areas and described by their frequency, bandwidth, and amplitude. We used the wide bandwidth of 3–12 Hz for our calculations to account for the gradual increase in the sensorimotor rhythm frequency from birth to adulthood [30].

To recognize sensorimotor rhythm and its focal slowing, we used the EEG data registered from central electrodes (C3, Cz, C4). We measured the minimal and maximal frequency of sensorimotor rhythm (SM_MinFr, SM_MaxFr), its index as the ratio between the duration of the EEG fragment with visually detected sensorimotor rhythm and the duration of the whole fragment (SM_Index), and the frequency rate as the ratio between high-frequency and low-frequency wave power (SM_FrR).

The calculations were performed after clinical inspection of the EEG by two separate clinical specialists when all 500 s EEG fragments were marked with the clinical EEG events including the appearance of the sensorimotor rhythm and its high and low frequencies.

To calculate the ratio between high-frequency waves and low-frequency waves of sensorimotor rhythm, we analyzed the EEG fragments with visually detected sensorimotor rhythm which were from 30 s to 359 s long. We calculated the power spectral density (PSD) using fast Fourier transformation (FFT) as density spectral array for the following spectral bands: 3–4 Hz, 4–5 Hz, 5–6 Hz, 6–7 Hz, 7–8 Hz, 8–9 Hz, 9–10 Hz 10–11 Hz and 11–12 Hz. For further analysis, we used log-transformed values.

For each subject, the individual minimal and maximal values of sensorimotor rhythm were taken into account. According to the median PSD, we identified the upper and lower frequency bands in sensorimotor spectra: we excluded the band of the median PSD and averaged all PSDs of bands with higher frequencies up to the maximal frequency band and all PSDs of lower bands up to the band including minimal frequency. The ratio was calculated as the mean PSD of higher bands to the mean PSD of lower bands. 

### 2.5. Statistical Analysis

We conducted a Spearman correlation to determine the strength and direction of a relationship between the severity of the disease (the total value of RSS), age and EEG data in a longitudinal part of our study. The multiple regression analysis was used to explain the impact of the aforementioned EEG parameters on the RSS.

The group analysis first identified group differences between RTT and TD using the Mann–Whitney test. Multiple regression was also used to estimate the impact of each EEG parameter on the total value of RSS in 32 patients. We then calculated Spearman partial correlation to evaluate the connections of EEG parameters that revealed significance in regression analysis with different scales in RSS taking age into account. Comparison of correlations for different EEG parameters was estimated for dependent samples according to [31] (p. 548).

Statistical analysis was performed using RStudio.

## 3. Results

### 3.1. Longitudinal Study: Patient U

#### 3.1.1. Demographic and Medical History

Patient U was born from a full-term pregnancy from healthy parents, who were 38 and 43 years old at the time of the girl’s birth. Her weight at birth was 3.920 g, length at birth was 53 cm and Apgar score was 9/10. Her first months of development were relatively normal with slight hypotonia and active babbling, although at 1 month of age she had periods of sundowning of the eyes and nystagmus/opsoclonus that was gone afterward. Head ultrasound at 3 months was completely normal. The concerns increased after several months when she did not start sitting or standing. At the age of 9 months, she was crawling only on her belly. At about the same time she acquired pincer grasp, but her voluntary hand movements were disturbed by hand soaking already after 5–7 months (by 1 year and 4 months). Thus, at the first EEG recording, when she was 1 year and 1 month old, her condition could be described as delayed motor development and might be considered the pre-regression or the beginning of the regression stage of the disorder (see Table 1 and Table 2).

The second EEG measures were at 1 y 7 m, and she already showed mild developmental regression—frequent mouthing of the hands, and no pincer grasp. No evident progress with motor skills was observed despite different therapies (physical therapy, speech therapy, massage, hippotherapy, swimming, etc.). She still could not sit independently or stand. She moved by crawling on her belly. Cognitive development was hard to assess. From birth, patient U liked listening to songs (preferring opera!) and poems, as well as attentively exploring/looking into books/pictures. She liked to be with people and watch them and liked to laugh with others; however, did not turn on her name. At about the same period she started to show episodes of shared attention.

At the time of the third measure at 2 y 6 m, she already had periods of hypoventilation and clear episodes of breath holding, especially when frightened. The motor activity decreased and she started crawling much less. So, this period can be also called regression.

EEG-recordings from 3 y 5 m correspond with the stationary stage of the disorder when no evident regression or progress was seen. At age 4 y 5 m, patient U was recruited in the clinical trial for the Evaluation of the Efficacy, Safety, and Tolerability of Sarizotan in Rett Syndrome with Respiratory Symptoms (STARS), (ClinicalTrials.gov Identifier: NCT02790034); thus, at her seventh EEG recordings at age 4 y 9 m, she was either on placebo or on this drug. From 5 y 1 m, she continued to participate in the open-label part of this trial that lasted until she was 5 y 10 m when the study was terminated. Thus, during the EEG recordings from eighth to eleventh, she was on sarizotan. During this period, she acquired new motor skills—she started to be able to stand with support and even walk with support. However, she was still not able to sit independently, and her crawling fully stopped. Just before sarizotan termination, when patient U was 5 y 9 m, new types of “strange” episodes occurred and persisted—grimaces that lasted a couple of minutes and usually ended with laughing. Here we should point out that while epilepsy was not diagnosed for patient U, parents reported rare episodes of “strange” behaviors such as freezing, nystagmus and trembling at awakenings during patient U’s first 6 years of life. Some of these events occurred during EEG recordings but did not correspond with clear epileptiform discharges. As these episodes come and go (sometimes several within a month, sometimes nothing for several months), no treatment for them was prescribed before this new “grimace” episode occurred. As the index of epileptiform activity also increased at this time, lamotrigine was prescribed at 50 mg/day with an increase to 100 mg/day starting from 5 y 10 m. At the time of the last EEG recording reported, at age 6 y 11 m, patient U was on lamotrigine and she still was not able to sit or walk without support and had very limited ways of communication with no words at all, but she had good eye contact (see Table 1).

A full assessment of patient U’s abilities was made at 4 y 6 m by a psychologist, speech therapist and physical therapist. The cognitive composite score on Bayley Scales of Infant Development (Third Edition) BSID-III was significantly delayed. She was not able to hold the object for more than 30 s as her grasp was very weak. She did not follow commands. She inconsistently tracked objects presented. Vineland Adaptive Behavior revealed a moderately low adaptive level in Socialization and Daily Living domain as well as low scores on Communication and Motor skills. She attempted to initiate an interaction by smiling and laughing in response to play. She did not display functional play with toys. According to the Preschool Language Scale (PLS-5), she displayed severe to profound delay in expressive and receptive language development. She did not follow an object that fell out of sight. She anticipated what would happen by smiling to peek-a-boo. She did not attempt to imitate facial expressions or movements. She displayed pleasure and displeasure sounds. Patient U did not display a representational gesture. Peabody Developmental Motor Scale 2 (PDMS-2) standard score was 2, composite score was 41, grasping score was 11 and visual motor integration score was 15. All these measures were below 3.6 standard deviations from the mean. Locomotion and Stationary raw scores were 12 and 20, respectively, corresponding to 1 standard score. Muscle weakness, scoliosis and overall delay in all her gross motor skills were registered. Oral peripheral examination revealed structure and functions to be within normal limits for speech and feeding purposes. Patient U presented poor saliva control (drooling). For eating, she needed food to be smashed.

She did not have epileptic seizure history before she was 6y 8m of age when atypical absences were detected during EEG registration. Overall, this phenotype is representative of the average RTT patient.

#### 3.1.2. EEG Evaluation

Table 2 represents all EEG parameters assessed in patient U during each session through her development and disease progression. Different types of epileptiform activity can be seen in her EEG: spikes, polyspikes, sharp waves with variable duration and frequency, spike–wave discharges, multiple spike–slow waves, and atypical sharp wave complexes and atypical absences (see Figure 1). While growing up, patient U demonstrated an increase in quantitative and qualitative epileptic EEG abnormalities (see Figure 1) that varied from single spike and wave discharges up to atypical absences. We would also like to show examples of patient U’s sensorimotor rhythm (Figure 2) that varied in frequency, amplitude and index through development. In central areas, we registered the SM with minimal frequency varying from 3.4 to 6.2 Hz. We found that during the disease progression, both SM_Fr and SM_Index decreased.

To examine the impact of the reported EEG abnormalities on RTT severity, we performed multiple regression analysis with total RSS values as a dependent parameter and all studied EEG parameters as regressors. The results of the regression indicated that the model explained 99% of the variance (adjusted R^2^ = 0.99). The model was a significant predictor of RTT severity (F (9.14) = 167.5, *p* = 0.00001). The regression equation had the following normalized beta-coefficients:RSS = 0.16 × GS_Fr + 0.53 × GS_Amp + 0.23 × Beta_Amp + (−0.24) × SM_MinFr + 0.43 × SM_MaxFr + 0.31 × SM_Index + (−0.61) × SM_FrR + (−0.1) × EEG_abnormalities

Several EEG parameters contributed significantly: GS_Amp (*p* = 0.003), Beta_Amp (*p* = 0.017), SM_MinFr (*p* = 0.002), SM_MaxFr (*p* = 0.014), SM_Index (*p* = 0.03) and SM_FrR (0.0006) contributed to the RTT severity. None of the parameters correlated with r > 0.8, which is why we did not have to consider multicollinearity here.

As age and RTT severity (RSS) were highly correlated (r = 0.8) for patient U, it was impossible to statistically differentiate the effect of age and the effect of disease progression using only this patient’s data. Thus, we continued the analysis with group data.

### 3.2. RTT Group Analysis and Comparison with TD Peers

#### 3.2.1. Demographic Features

The average age of 32 girls with Rett syndrome was 8.46 ± 4.15, range 1.9–17.1. The average age of the symptoms’ onset was 17 ± 5 months, range 8–30. This group was compared with 41 typically developed girls of similar age (average age 9.1 ± 3.46, range 2.58–17.98).

#### 3.2.2. Medical History Features

In the clinical group, seven cases (22%) had epileptic seizure history. Two of these cases had generalized tonic–clonic convulsions, two of them also had atypical absences, one patient had tonic–clonic convulsions and two girls had myoclonus. Two of the patients used valproic acid, one of them used carbamazepine, one of them used lamotrigine, and others used two or more therapeutic agents. Participants in the control group did not have a history of epileptiform activity or any clinically relevant EEG abnormalities.

#### 3.2.3. Epileptiform and Paroxysmal Activity

The visual analysis of epileptiform activity identifies three types of epileptiform activity registered in the RTT group and partly in controls. Benign sporadic wave discharges were found in both groups of children. In the control group of children, it was registered in 40% of cases; the total duration did not exceed 1% of the total EEG recording. These benign spikes or discharges in girls with Rett syndrome were found in 97% of cases and had an average duration of 3.1% of the analyzed EEG fragment. 

Atypical epileptiform discharges manifesting with secondary generalized spike–slow wave discharges or spike–wave discharges in the control group of children were registered in 6.1% of cases, and the average duration was 0.2% of the total EEG recording. This type of epileptiform activity in the RTT group was registered in 97% of cases, had an average duration of 7.9% of the analyzed EEG fragment and was localized in frontal areas. Finally, 29% of girls with Rett syndrome demonstrated the atypical and typical spike–slow wave secondary generalized discharges which were localized in the frontal area and had a 15.7% mean duration. We did not reveal this type of discharge in the TD group.

#### 3.2.4. EEG Spectral Changes

Spectral characteristics of spontaneous EEG differed between TD and RTT in almost all measured parameters (Table 3). In RTT, general background activity was of lower frequency (GS_Fr) and higher amplitude (GS_Amp) as compared to TD peers (Mann–Whitney test for GS_Fr: Z = −3.98, *p* = 0.0001; Mann–Whitney test for GS_Amp: Z = 4.20, *p* = 0.0001; see Table 4). Sensorimotor rhythm was also of higher amplitude (SM_amp) and lower frequency (SM_MinFr, SM_MaxFr), with its representation in EEG (SM_index) decreased in RTT (Mann–Whitney test for SM_amp: Z = 3.346, *p* = 0.0007; Mann–Whitney test for SM_MinFr: Z = −4.608, *p* < 0.0001; Mann–Whitney test for SM_MaxFr: Z = −2.37, *p* = 0.017; Mann–Whitney test for SM_index: Z = −2.69, *p* = 0.007; see Table 4). Our new index, SM frequency rate (SM_FrR), that shows the predominance of high vs. low SM rhythm in EEG was drastically decreased in RTT, pointing to the shift of SM towards lower values even within individual variability of this parameter that takes into account the decreased individual median values (Mann–Whitney test for SM_FrR: Z = −5.54, *p* < 0.0001; see Table 3). At the group level, such a decrease in SM_FrR is linked to a more than 2-fold decrease in the SM_MinFr which is about 3 Hz in RTT but 6 Hz in TD. The difference between waves with the lowest and highest frequencies was from 3.2 to 9.9 Hz in RTT, which is not typical for TD children, who had average lowest and highest frequencies of 6.4 to 10.7, respectively. Only beta rhythm frequency was similar between groups (see Table 4, Figure 3).

Multiple regression was conducted to investigate whether EEG parameters could significantly predict RSS scores. The results of the regression indicated that the model explained 64% of the variance (with adjusted R^2^ = 0.64). The model was a significant predictor of RTT severity (F (8.23) = 7.967, *p* = 0.00004). The regression equation had the following normalized beta-coefficients:RSS = −0.08 × GS_Fr + 0.54 × GS_Amp + 0.17 × Beta_Amp + (−0.15) × SM_MinFr + 0.2 × SM_MaxFr + (−0.06) × SM_Index + (−0.5) × SM_FrR + (−0.16) × EEG_abnormalities

Two EEG parameters contributed significantly: GS_Amp (*p* = 0.0005) and SM_FrR (*p* = 0.0007) (see Figure 3). None of the EEG parameters correlated with r > 0.8, which is why we did not have to consider multicollinearity here. These two parameters also differed significantly between RTT and TD groups (Mann–Whitney test for GS_amp: Z = 4.20, *p* = 0.0001; Mann–Whitney test for SM_FrR: Z = −5.54, *p* < 0.0001; Table 4) and significantly correlated with the total score of RSS in patients (GS_amp: r = 0.68, *p* < 0.0001; SM_FrR: r = −0.61, *p* = 0.0002, Spearman’s correlation, see Figure 4). The higher the amplitude of general background slowing is and the lower the sensorimotor frequency rate is, the more severe the RTT phenotype observed is.

We also examined how these two parameters were connected with distinct RSS scales and calculated partial correlation taking age as a covariate, as it was significantly correlated with GS_amp (r = 0.47, *p* = 0.006) but not with SM_FrR (r = −0.16, *p* = 0.38). GS_amp correlated significantly with the following RSS scales: Walk (r = 0.47, *p* = 0.006), Hand use (r = 0.5, *p* = 0.003), Speech (r = 0.45, *p* = 0.01) and Sleep (r = 0.35, *p* = 0.05). SM_FrR, on the other hand, correlated with Seizures (r = −0.41, *p* = 0.02), Breathing (r = −0.35, *p* = 0.05), Walk (r = −0.49, *p* = 0.005) and Hand use (r = −0.4, *p* = 0.02) (see also Table 5). Noteworthily, correlation coefficients between RTT subscales and EEG parameters of interest reached statistical difference only for the Breathing irregularities subscale which showed a significantly higher correlation with sensorimotor frequency rate than with amplitude of general background slowing. Thus, we speculate that the amplitude of general slowing and sensorimotor frequency rate is linked with slightly different profiles of RTT manifestations.

## 4. Discussion

Our study is one of the largest EEG studies of patients with RTT. In line with previous research, we showed substantial EEG abnormalities in patients with RTT as compared to their typically developing peers. Those included an increased percentage of epileptiform activity, increased amplitude and decreased frequency of general background slowing, and decreased frequency and prevalence of sensorimotor activity in resting EEG. Extending previous findings, we showed that some of the EEG features obtained from clinical EEG are significantly correlated with the severity of RTT symptoms and the course of the disease progression both in a longitudinal case study and at the group level. Those were the amplitude of general background slowing and a new measure of sensorimotor rhythm individual variability—sensorimotor frequency rate (SM_FrR). Noteworthily, these two neurophysiological indexes were related to slightly different profiles of RTT symptoms. Below we discuss these results in detail.

We start the discussion with EEG epileptiform activity. In line with previous research, we confirm that RTT exhibits a high prevalence of benign epileptiform activity and episodic peak–wave or slow spike–wave complexes, which were not accompanied by clinical events—97% of patients with RTT in our study. At the same time, electroencephalographic seizures were registered in substantially smaller numbers of cases—29%. In spite of the prevalence of EEG abnormalities in RTT, the total index of detected EEG paroxysmal abnormalities did not correlate with RSS symptoms, indicating that EEG abnormalities are not directly related to the RSS severity and rather represent the comorbid factor potentially linked to genetic profile [32,33,34,35]. Epileptiform activity is unlikely the causal factor for RTT development.

Another measure that is frequently used in clinical EEG is general slowing in background activity. The predominance of slow-wave activity in girls with RTT was frequently reported (for a review, see [11]). Noteworthily, it was linked to behavioral symptoms, such as deteriorated speech and language functions [36], decreased scores on the Mullen scale of early learning [37] and worse cognitive functioning [38]. The general background slowing of EEG that could indicate diffuse cerebral dysfunction showed an association with RSS total score in our study. Previous studies showed that different etiologies may provoke general EEG slowing, including hydrocephalus, neurodegenerative disorders, metabolic encephalopathy, or even structural lesions involving diencephalic structures or the brainstem [29,39]. While general slowing indicated diffuse, not specific cerebral dysfunction, it may also be taken as a marker of neurodevelopmental disorders, such as autism spectrum disorders [40], developmental language disorder [41], attention deficit hyperactivity disorder [42], GABRB2-associated neurodevelopmental disorders [43] and other neurodevelopmental syndromes [44]. In our study it was associated with particular RTT severity subscales, such as Walk, Hand use, Speech, and Sleep, also pointing to the link of more pronounced generalized slowing with substantial language and motor delay. Noteworthily, the amplitude of general background slowing was associated with RTT severity not only at the group level, but also for an RTT patient that was followed longitudinally, indicating that this measure has a potential for application at the individual level.

The abnormality in sensorimotor rhythm, in particular its substantial slowing, was also previously reported in RTT (for a review, see [11]). Our study confirmed this finding. In addition, we introduce the new measure of sensorimotor rhythm—SM_FrR, which reflects the variability range of frequencies of this rhythm. In particular, we found that during EEG recording, the frequency of sensorimotor rhythm in RTT could change from high to low frequency and vice versa while maintaining the same topography. To assess these alterations, we calculated the ratio between high-frequency and low-frequency waves and called it SM_FrR. This SM_FrR parameter was significantly lower in RTT than TD, pointing to the prevalence of lower frequencies of sensorimotor rhythms even when considered around individual SM frequency median. Moreover, this parameter significantly predicted the RTT severity and in particular was associated with Breathing irregularity, Seizures, Walk and Hand use subscales of RSS. While associations with Walk and Hand use subscales were shared with general background slowing, breathing irregularities correlated only with SM_FrR, pointing to partially different profiles of symptoms related to these two relevant to RTT EEG measures. According to previous findings, EEG signals and brain function were extremely sensitive to the lack of O2 supplied by breathing abnormalities [45,46]. It is well known that hyperventilation provocation tests could induce EEG changes from disorganization of basic rhythm and decreases in the alpha frequency range up to generalized high-amplitude slow synchronous waves even in persons without epileptic activity or any neurological symptoms [47,48]. At the same time, the progressive hypercapnia with iso-hypoxia was associated with changes in the ratio of slow-wave activity to alpha rhythm [49]. Thus, the SM_FrR could be sensitive to one of the most specific respiratory symptoms of Rett syndrome that could be causative of other RTT symptoms, such as seizures, and motor dysfunction. While this has to be confirmed in the other studies that could track the onset of breathing irregularities and their severity in relation to other RTT symptoms, our study indicates that intervention targeting the normalization of breath might have great potential for the amelioration of RTT severity. Our longitudinal case report also supports this idea as treatment with sarizotan that aimed to improve breathing activity led to slight progress in motor functions—the girl acquired standing skill during treatment.

The absence of correlation of some abnormal EEG parameters with RTT progression in our study might indicate that they are indeed not related to RTT symptoms representing some general neurophysiological deficits. At the same time, their contribution to RTT symptoms can be relatively small to be caught by our study, or our measures of RTT severity might not possess the necessary sensitivity and wideness.

## 5. Conclusions

In conclusion, our study combined a longitudinal case report and group-level analysis to examine the association of clinical/qualitative and experimental/quantitative EEG measures with the severity of Rett syndrome. We confirmed a high prevalence of EEG paroxysmal activity in RTT but did not find any evidence for the association of this parameter with RTT progression. At the same time, the amplitude of general background slowing as well as our newly introduced measure of sensorimotor rhythm frequency variability, SM_FrR, showed a relationship with RTT severity both at the group level and in a longitudinal case: the more severe RTT symptoms were associated with a larger amplitude of general background slowing and larger variability of sensorimotor rhythm due to predominance of low-frequency activity.

## Figures and Tables

**Figure 1 jpm-12-01973-f001:**
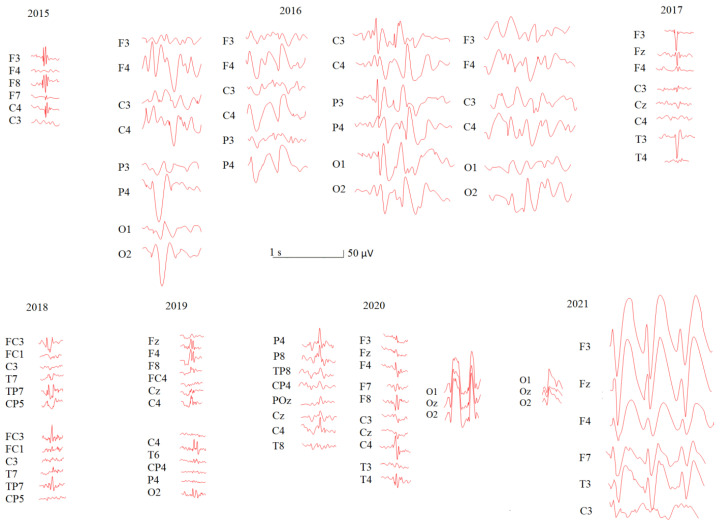
Epileptic EEG abnormalities in patient U during the course of the disease. The EEG recordings marked by stars in Table 2 were used for the visualization.

**Figure 2 jpm-12-01973-f002:**
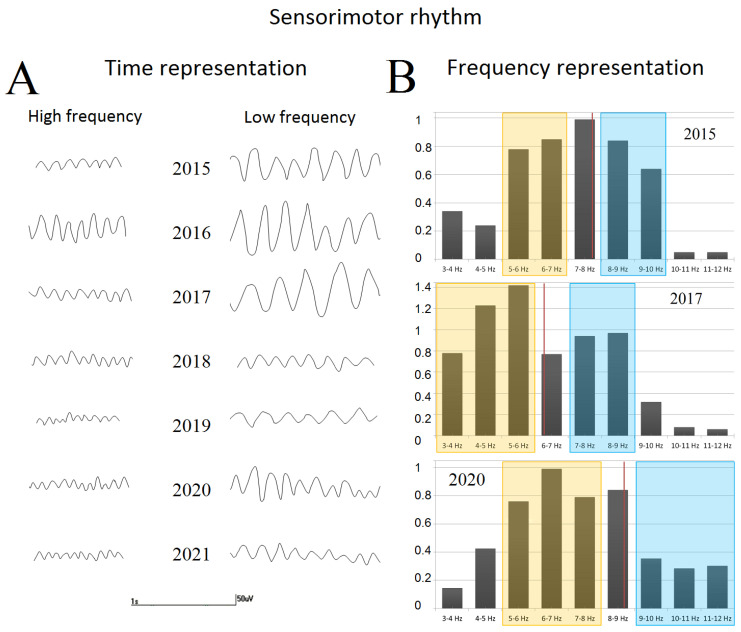
(**A**) The examples of fast and slow waves of sensorimotor (µ) rhythm (Cz). The EEG recordings marked by stars in Table 2 were used for the visualization. (**B**) The visualization of frequency rate of sensorimotor rhythm (SM_FrR) calculations for the EEG recording of patient U registered on 21 August 2015, 18 January 2017 and 20 March 2021; red vertical line corresponds to the median PSD that divides PSD into high and low values, marked respectively by blue and orange boxes. The averages over these boxes were used to calculate SM_FrR. x—frequency bands, y—PSD.

**Figure 3 jpm-12-01973-f003:**
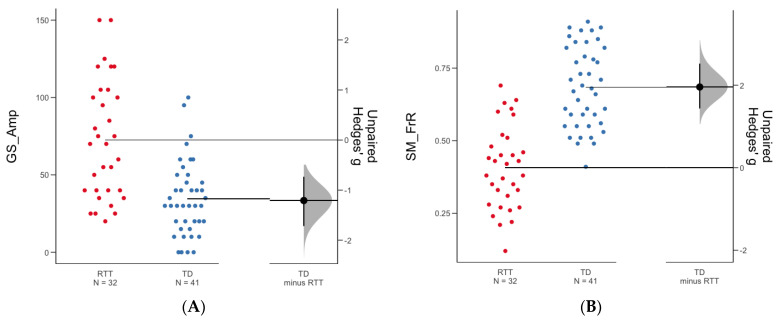
The mean difference between RTT and TD is shown in the above Gardner–Altman estimation plot for two parameters: (**A**) frequency rate of sensorimotor rhythm (SM_FrR); (**B**) amplitude of general slowing (GS_Amp). Both groups (children with Rett syndrome, RTT, and typically developed group, TD) are plotted on the left axes; the Hedges’ g effect size is plotted on floating axes on the right.

**Figure 4 jpm-12-01973-f004:**
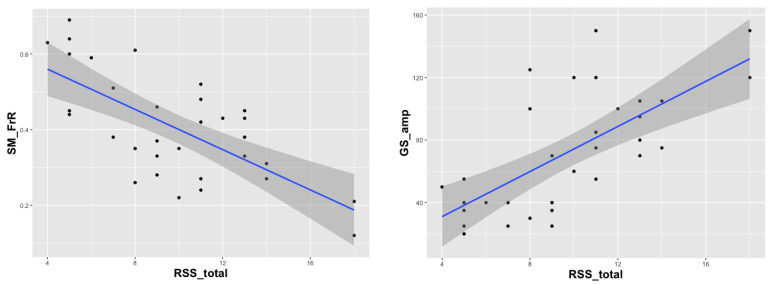
Scatterplots of RSS total scores against the frequency rate of sensorimotor rhythm frequency rate (SM_FrR, r = −0.61, *p* = 0.0002) and the amplitude of the general slowing of sensorimotor rhythm (GS_amp, r = 0.68, *p* < 0.0001) for RTT patients (Spearman’s correlation).

**Table 1 jpm-12-01973-t001:** Behavioral phenotype (Rett Syndrome Severity Subscales (RSSSs)) of patient U for corresponding EEG recordings. 0—no deficit, 3—severely affected with exact characteristics specific to particular subscale.

Age at EEG Registration	Medications	Rett Syndrome Severity Subscales (Scores)								RTT Stages
		Total	Seizures	Breath	Scoliosis	Walk	Hand Use	Speech	Sleep	
1 y 1 m		8	0	0	0	3	1	3	1	I
1 y 7 m		8	0	0	0	3	1	3	1	II
2 y 6 m		12	0	2	1	3	2	3	1	II
3 y 5 m		12	0	2	1	3	2	3	1	III
3 y 10 m		12	0	2	1	3	2	3	1	III
4 y 4 m		12	0	2	1	3	2	3	1	III
4 y 9 m	Sarizotan or placebo	12	0	2	1	3	2	3	1	III
5 y 2 m	Sarizotan open-label	12	0	2	1	3	2	3	1	III
5 y 3 m	Sarizotan open-label	12	0	2	1	3	2	3	1	III
5 y 4 m	Sarizotan open-label	12	0	2	1	3	2	3	1	III
5 y 7 m	Sarizotan open-label	12	0	2	1	3	2	3	1	III
6 y 1 m		12	0	2	1	3	2	3	1	III
6 y 8 m		13	1	2	1	3	2	3	1	III
6 y 11 m	Lamotrigine	13	1	2	1	3	2	3	1	III

**Table 2 jpm-12-01973-t002:** EEG data of patient U relevant for 14 EEG recordings for several parameters: general slowing (GS), sensorimotor rhythm (SM), beta rhythm (*β*) and epileptiform activity. Max—maximal, Min—minimal, Fr—frequency, FrR—frequency rate as the ratio between high-frequency and low-frequency waves, Amp—amplitude, Index—the percentage of EEG parameters to the whole analyzed EEG fragments.

EEG Parameters									
	GS		SM (*µ*)				*β* (*µ*)	Epileptiform	
Age at EEG Registration	Fr	Amp	Amp	MinFr	MaxFr	Index	FrR	Amp	Index
1 y 1 m *	1.3	35	140	5.2	8.6	63.9	1.17	25	11.6
1 y 7 m *	1.5	30	195	4.6	7.9	54.7	0.86	20	25.7
2 y 6 m *	1.5	85	185	3.9	8.7	46.8	0.37	20	20.2
3 y 5m	1.9	90	145	5.2	9.5	39.4	0.35	20	32.1
3 y 10 m *	2.3	95	65	6.1	10.1	34.4	0.34	20	38.5
4 y 4 m	2.4	90	70	5.5	10.5	35.25	0.42	20	29.4
4 y 9 m	2.3	85	60	4.3	10.9	29.25	0.38	15	21.5
5 y 2 m *	2.1	100	68	4.4	11.8	33.8	0.66	15	18.6
5 y 3 m	1.9	90	72	4.9	11.3	41.8	0.45	15	14.9
5 y 4 m	2.7	95	88	6.8	11.8	44.5	0.55	15	16.7
5 y 7 m	2.4	95	94	5.1	11.2	38.2	0.41	15	19.9
6 y 1 m *	2.2	95	112	6.6	11.4	36.1	0.38	15	12.2
6 y 8 m *	2.6	125	55	6.1	11.8	18.1	0.24	15	62.5
6 y 11 m	2.5	130	65	5.5	11.7	17.6	0.22	12	44.1

*—EEG recordings which are used in Figure 1 and Figure 2.

**Table 3 jpm-12-01973-t003:** Descriptive statistics of demographic and medical RTT and TD groups. RSS scores are presented only for patients (0—no deficit, 3—severely affected with exact characteristics specific to particular subscale).

		Valid *N*	Mean	Median	Std. Dev.	Std. Error
Age TD		41	9.106	9.04	3.457	0.54
Age RTT		32	8.46	8.08	4.15	0.73
age of regression, RTT (month)		32	17.45	17.5	5.6	1.19
** *RSS* **	Seizures	31	0.81	0	0.98	0.18
	Breath irregularities	31	1.19	1	1.05	0.19
	Scoliosis	30	1	1	1.05	0.19
	Walk	31	1.94	2	1.12	0.2
	Hand use	32	1.91	2	1.03	0.18
	Speech	32	2.53	3	0.51	0.09
	Sleep	31	0.58	0	0.76	0.14
	Total	32	9.75	9.5	3.65	0.65

**Table 4 jpm-12-01973-t004:** EEG parameters in patients with Rett syndrome (RTT) and typically developed (TD) children. Following EEG parameters are presented: general slowing (GS), sensorimotor rhythm (SM), beta rhythm and EEG abnormalities. Max—maximal, Min—minimal, Fr—frequency, FrR—frequency rate as the ratio between high-frequency and low-frequency waves, Amp—amplitude, Index—the percentage of EEG parameters to the whole analyzed EEG fragments.

RTT (*n* = 32)			TD (*n* = 41)		
	Mean ± STD	Median	Mean ± STD	Median	Mann–Whitney U test
GS_Fr	2.55 ± 0.62	2.40	3.23 ± 0.65	3.2	Z = −3.98, *p* = 0.0001
Gs_Amp	72.50 ± 38.33	70.00	35.24 ± 26.31	30	Z = 4.20, *p* = 0.0001
SM_Amp	108.81 ± 49.62	107.50	73.17 ± 24.51	65	Z = 3.346, *p* = 0.0007
SM_MinFr	3.16 ± 1.34	3.55	6.42 ± 1.39	6.3	Z = −4.608, *p* < 0.0001
SM_MaxFr	9.89 ± 1.86	10.20	10.71 ± 1.19	10.91	Z = −2.37, *p* = 0.017
SM_Index	18.95 ± 20.06	9.88	25.95 ± 14.48	23.4	Z = −2.69, *p* = 0.007
SM_FrR	0.41 ± 0.14	0.40	0.66 ± 0.14	0.665	Z = −5.54, *p* < 0.0001
Beta_Amp	20.13 ± 7.89	19.75	16.63 ± 4.74	15	Z = 1.67, *p* = 0.084
Epileptiform activity	8.78 ± 13.53	5.13	0.86 ± 1.68	0.11	Z = 5.63, *p* < 0.0001

**Table 5 jpm-12-01973-t005:** Partial correlation of age and RSS scales with mean amplitude of general slowing (GS_amp) and frequency rate of sensorimotor rhythm (SM_FrR). Comparison of correlations is estimated for dependent samples according to [31]; single-sided testing. Significant differences are presented in bold.

RSS Subscales	Partial Correlation with GS_amp	Partial Correlation with SM_FrR	Comparison of GS and SM Correlations
Seizures	r = 0.19, *p* = 0.3	**r = −0.41, *p* = 0.02**	*p* = 0.12
Breath irregularities	r = −0.09, *p* = 0.6	**r = −0.35, *p* = 0.05**	***p* = 0.04 **
Scoliosis	r = 0.3, *p* = 0.09	r = −0.19, *p* = 0.3	*p* = 0.28
Walk	**r = 0.47, *p* = 0.006**	**r = −0.49, *p* = 0.005**	*p* = 0.45
Hand use	**r = 0.5, *p* = 0.003**	**r = −0.4, *p* = 0.02**	*p* = 0.28
Speech	**r = 0.45, *p* = 0.01**	r = −0.3, *p* = 0.1	*p* = 0.21
Sleep	**r = 0.35, *p* = 0.05**	r = −0.2, *p* = 0.2	*p* = 0.21
Total	r = 0.58, *p* = 0.0005	r = −0.62, *p* = 0.0002	*p* = 0.39

## Data Availability

Not applicable.

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
