# Peer review of "Clinical EEG of Rett Syndrome: Group Analysis Supplemented with Longitudinal Case Report"

_jpm, 2022, doi:10.3390/jpm12121973_

Round 1

Reviewer 1 Report

This study presents findings of the EEG characteristics and disease progression in one Rett syndrome patient. Additionally, EEG characteristics of another cohort of patients is compared with those in the normal kids.

My only concern is that a lot of data are presented in tables 1 and 2. Perhaps splitting these data in multiple tables will help highlight the EEG characteristics.

Since the authors did not find a correlation between EEG characteristics and disease progression some discussion as to the reasons for this negative findings will be helpful.

Author Response

We would like to thank the reviewers for their evaluation of our ms. and their informative comments. We tried to take all of them into account and corrected the text.

Please find our detailed responses to your comments below:

This study presents findings of the EEG characteristics and disease progression in one Rett syndrome patient. Additionally, EEG characteristics of another cohort of patients is compared with those in the normal kids.

  • My only concern is that a lot of data are presented in tables 1 and 2. Perhaps splitting these data in multiple tables will help highlight the EEG characteristics.

We divided Table 1 into two as suggested..

  • Since the authors did not find a correlation between EEG characteristics and disease progression some discussion as to the reasons for these negative findings will be helpful.

We found correlation for mean amplitude of general slowing (GS_amp) and frequency rate of sensorimotor rhythm (SM_amp) and RSS scores and we discuss in the Discussion section (lines 521-527). Unfortunately due to some technical reasons this figure was absent in the ms. version the reviewer received. At the same time, some of the EEG measures did not show the correlation with RSS severity, e,i. the degree of epileptiform abnormalities, beta amplitude, frequency of generalized slowing, some parameters of sensory-motor rhythms. We now added our comments for the absence of these correlations.

“The absence of correlation of some abnormal EEG parameters with RTT progression in our study might indicate that they are indeed not related to RTT symptoms representing some general neurophysiological deficits. At the same time, their contribution to RTT symptoms can be relatively small to be caught by our study or our measures of RTT severity might not possess necessary sensitivity and wideness.”

Reviewer 2 Report

I reviewed carefully the manuscript titled Clinical EEG of Rett syndrome: group analysis supplemented with longitudinal case-report, however, major and minor modifications are needed:

Major:

In 2.2. EEG registration is not mentioned the conditions of performed: awake or sleep? and technical details: as duration of EEG recording? filters? and use of Notch?

Why included benign variant with pathological patterns?

Is unclear the term pseudo generalization, I recommend use the international EEG terminology in the manuscript and methods.

3 Beta-rhythm express the unit of measurement are needed.

Minor:

In line 92: what is the city and country of fabrication of the device

In line 189 correct Manna-Whitney test

In Table 2 add the unit of measurement for example, years old, including the unclear terms walk and scoliosis isolate in the table, for example walk score, presence of scoliosis, etc.,

In line 505 epyleptic activity

In The Discussion section line 446, decreased frequency of general background slowing is redundant.

Author Response

We would like to thank the reviewers for their evaluation of our ms. and their informative comments. We tried to take all of them into account and corrected the text.

Please find our detailed responses to your comments below:

Major:

  1. In 2.2. EEG registration is not mentioned the conditions of performed: awake or sleep? and technical details: as duration of EEG recording? filters? and use of Notch? 

We included this information to the text (lines 94-98).

  1. Why include benign variants with pathological patterns?

We added argumentation to the text (lines 127-136):

“Benign variants were included for several reasons. First, we believe that the epileptiform activity can be easily misclassified especially in children with atypical development [26–28]. Second, the EEG abnormalities including benign variants of epileptiform activity could be a sign of both brain immaturity and brain pathology [27] and we suggested that we need take into account this factor when comparing RTT children with their typical peers.”

  1. Is unclear the term pseudo generalization, I recommend use the international EEG terminology in the manuscript and methods. 

Term “pseudo generalization” is widely used in national clinical EEG analysis, and we did not realize that it is not that well-known internationally. Currently we changed to the term “secondary generalized” (Mironov, 2017) that is more conventional.

  1. Beta-rhythm express the unit of measurement are needed. 

We added units for beta-rhythm in Table 1.

  1. Minor:

All minor corrections were implemented.

In line 92: what is the city and country of fabrication of the device —  done

In line 189 correct Manna-Whitney test —  done

In Table 2 add the unit of measurement for example, years old, including the unclear terms walk and scoliosis isolate in the table, for example walk score, presence of scoliosis, etc.,

Currently we indicated that we used the RSS scores for different subscales in the tables more clearly (need to be done when splitting the table 1)

In line 505 epyleptic activity —  done

In The Discussion section line 446, decreased frequency of general background slowing is redundant.

Here we wanted to emphasize two different aspects of general background slowing - its amplitude and frequency that we wrote “increased amplitude and decreased frequency of general background slowing”.

We also noticed that one figure (Figure 4) was absent in the original manuscript, however mentioned in the text itself. Now we have corrected this mistake.